# Do LLMs have Consistent Values?

**Naama Rozen**
Tel-Aviv University
naamarozen240@gmail.com

**Liat Bezalel**
Tel-Aviv University
liatbezalel@mail.tau.ac.il

**Gal Elidan**
Google Research
Hebrew University
elidan@google.com

**Amir Globerson**
Google Research
Tel-Aviv University
amirg@google.com

**Ella Daniel**
Tel-Aviv University
della@tauex.tau.ac.il

## Abstract

Large Language Models (LLM) technology is rapidly advancing towards human-like dialogue. Values are fundamental drivers of human behavior, yet research on the values expressed in LLM-generated text remains limited. While prior work has begun to explore value ranking in LLMs, the crucial aspect of value correlation – the interrelationship and consistency between different values – has been largely unexamined. Drawing on established psychological theories of human value structure, this paper investigates whether LLMs exhibit human-like value correlations within a single session, reflecting a coherent "persona". Our findings reveal that standard prompting methods fail to produce human-consistent value correlations. However, we demonstrate that a novel prompting strategy (referred to as "Value Anchoring"), significantly improves the alignment of LLM value correlations with human data. Furthermore, we analyze the mechanism by which Value Anchoring achieves this effect. These results not only deepen our understanding of value representation in LLMs but also introduce new methodologies for evaluating consistency and human-likeness in LLM responses, highlighting the importance of explicit value prompting for generating human-aligned outputs.

## 1 Introduction

A central objective in the development of Large Language Models (LLMs) is to create agents capable of "human-like" communication. While LLMs have a remarkable ability to generate fluent text across diverse tasks, human communication extends beyond mere fluency to encompass internal consistency and complex psychological characteristics. One fundamental aspect of this internal consistency is the human value system. Values, as basic motivations guiding perceptions and behaviors, exhibit a structured organization, with certain values being inherently correlated or conflicting within individuals (Sagiv and Schwartz, 2022). This raises several key questions: during a single conversation with an LLM, does the "LLM-persona" resemble a single human in terms of the way values are manifested? Furthermore, across multiple conversations, can LLMs produce multiple personas that resemble a population of humans? And if this is indeed possible, how can such personas be elicited to best resemble psychological characteristics observed in human populations?

This question has only recently begun to be addressed. For example, Aher et al. (2023) show how probing LLMs with different names leads to variability which in some cases agrees with that of human populations. Studies like Fischer et al. (2023) and Lindahl and Saeid (2023) have investigated value rankings in LLM outputs. However, our focus is different: we ask whether an LLM in a single conversation can exhibit a psychological characteristic profile that resembles human patterns. This is a highly challenging question, as it requires analyzing complete conversations to evaluate whether they could conceivably have been generated by a single individual.

To establish a quantitative framework for evaluation, we turn to the well-established field of value psychology. Specifically, we aim to quantify the values that LLM responses are aligned with, and whether these align with the value hierarchy and structure observed in humans. Values are basic

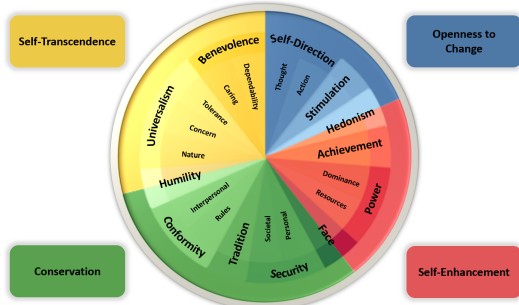

Figure 1: Circular motivational continuum of 19 values in the refined value theory. Source: Schwartz et al. (2012). A value aligns with values that are adjacent on the circle and conflicts with those opposite to it. For example, self direction aligns with stimulation, and both conflict with conformity.

motivations that play a foundational role in psychology, influencing perceptions and behaviors across various domains (Sagiv and Schwartz, 2022; Sagiv et al., 2017), and representing fundamental aspects of human personality (Roberts and Yoon, 2022). Research has consistently demonstrated their enduring influence over behavior across time and contexts (Sagiv et al., 2017).

The Theory of Basic Human Values (Schwartz, 1992) provides a robust framework for the investigation of values in human personality. It identifies 19 core values categorized by motivational goals (Schwartz, 2012). These values can be mapped onto a two-dimensional structure: conservation versus openness to change, and self-enhancement versus self-transcendence. The theory describes interrelations among values, suggesting that certain motivational goals are compatible while others conflict. For example, pursuing independence and creativity (self-direction) aligns naturally with seeking change and variability (stimulation) but conflicts with maintaining the status quo (conformity). These interrelations are expressed in consistent statistical associations between the value priorities of individuals, expressed in a circumplex structure. See Figure 1 for the theorized circle of values, where values that naturally align are next to each other. The same values and their associations were repeatedly identified in hundreds of samples, across multiple cultures and age groups (Sagiv and Schwartz, 2022; Sagiv et al., 2017)

Our key question is therefore whether LLM responses demonstrate the same statistical behavior observed in humans with respect to both value-ranking and value-correlations. Note that the question of value-correlations is of particular interest, because it provides a benchmark for the extent to which responses of an LLM demonstrate a coherent "persona". For example, while it is possible for a person to give a high score to Power Dominance, that person is unlikely to give a high score to Benevolence, since these are contradicting values.

To study this question quantitatively, we present LLMs with a well validated value questionnaire (the Portrait Value Questionnaire—Revised – PVQ-RR– from Schwartz (2017)), and prompt them to answer all the questions in a single session (i.e., in the same context window). We then analyze the provided answers, putting specific emphasis on the correlation between answers in the same session.

We analyze two recent LLMs: GPT-4-0314 and Gemini 1.0 Pro, as well as four open models: Llama 3.1 8B, Llama 3.1 70B, Gemma 2 9B, and Gemma 2 27B.[1] Our results show that standard prompting of LLMs *does not* result in a population of human-like personas. We go on to explore prompting the LLMs with other prompts that provide additional information about the LLM persona. In particular we consider previously used prompting of names (Aher et al., 2023) and persona descriptions. In addition, we consider a novel prompt which we refer to as a "Value Anchor", which instructs the language model to answer as a person emphasizing a given value. We find that with these prompts, and in particular with the Value Anchor prompt, the overall first and second order statistics of the LLM responses closely mirror those of human subjects. We furthermore provide an explanation for how this statistical behavior comes about. We include six datasets comprising 300 personas each, generated by the models. In conclusion, our results demonstrate the utility of using psychological theory to evaluate the consistency of personas generated by LLMs.

---

[1]Our analysis also included the earlier GPT-3.5-turbo and Palm2, which produced qualitatively similar results, not reported here for brevity.

## 2 RELATED WORK

**Values in LLMs:** Our work builds upon the well-established Schwartz Theory of Personal Values (Sagiv and Schwartz, 2022), a framework widely accepted in personality psychology for understanding human motivations and behavior. This theory posits that values are fundamental, abstract goals that guide individual judgments and actions (Schwartz, 1992; 2012). Although there is variability between individuals in their prioritization of values, there are values that tend to be ranked as more important than others across cultures and samples. Those suggest there are underlying principles that give rise to value hierarchies. The similarity in value importance across cultures is referred to as the universal value hierarchy (Schwartz and Bardi, 2001; Schwartz and Cieciuch, 2022). A second core aspect of the Schwartz Theory of Personal Values, posits that values are structured, with individuals likely to report similar importance of compatible motivations, and varying importance of conflicting motivations – a pattern of associations consistently observed across diverse human populations (Pakizeh et al., 2007; Skimina et al., 2021a). These robust and cross-cultural aspects of the value theory makes it a particularly useful lens through which to examine the coherence of value profiles in Large Language Models (LLMs). While the exploration of values in LLMs is still a relatively new area, as highlighted in a recent survey by Ma et al. (2024), initial studies have begun to emerge, each taking different approaches to understand how these models represent and express values. Many early studies adopted what can be termed an "LLMs as individuals" perspective, treating these models as singular entities capable of comprehending and expressing human values. Fischer et al. (2023) probed ChatGPT's basic comprehension of values through value-laden prompts, while Lindahl and Saeid (2023) benchmarked ChatGPT's value expressions against the World Value Survey's cross-cultural dataset. Other researchers explored specific dimensions: Miotto et al. (2022) investigated how temperature settings influenced value expression, Scherrer et al. (2023) examined moral positions, and Hadar-Shoval et al. (2024) analyzed value-like constructs. Durmus et al. (2023) contributed by creating a dataset to evaluate LLM representations against cross-national survey data. A shift in perspective came with Kovač et al. (2023), who challenged the "LLMs as individuals" assumption by demonstrating the significant impact of context on ChatGPT's value expressions. As a result, they suggested that LLMs do not represent a singular value, but can produce a myriad of values. Subsequent work (Aher et al., 2023; Argyle et al., 2023; Kang et al., 2023) expanded on this insight, exploring various prompting techniques to elicit and manipulate value-related responses, from demographic information to name-based prompts. Previous research has focused on aspects such as value stability (Kovač et al., 2024) and consistency (Lee et al., 2024; Wang et al., 2024). Despite these advances, much remains to be understood. Previous work that created multiple values using prompting techniques had hardly approached the question of the nature of those created values. Specifically, they did not investigate the internal coherence of a value persona constructed by an LLM during a single session. As a result, they could not investigate the value hierarchies of LLM created personas, and the nature of the interrelations between the values, in line with existing information about human values. Our methodology advances this line of research by examining structured relationships between values within LLMs. Our findings reveal that LLMs exhibit value correlations that mirror Schwartz's circular model, suggesting these models have internalized not just individual values, but also their theoretical interrelationships.

**Prompting LLMs:** Researchers have explored diverse approaches to elicit individual characteristics from LLMs through prompting (Liu et al., 2023). These methods include presenting specific scenarios (Hadar-Shoval et al., 2023), administering questionnaires (Jiang et al., 2023), simulating social identities and expertise (Salewski et al., 2024), using gendered and ethnic names (Aher et al., 2023), and incorporating demographic information (Argyle et al., 2023). More sophisticated techniques involve designated personas (Safdari et al., 2023) and RLHF (Li et al., 2023) to instill distinct personality traits. However, as Zheng et al. (2024) demonstrate, LLM outputs are highly sensitive to subtle variations in prompting, highlighting the importance of careful prompt design and rigorous evaluation across different prompting strategies. While this body of research is substantial, there is insufficient systematic comparison of these various prompting techniques, specifically with regards to their ability to simulate consistent psychological characteristics.

**Temperature in LLMs:** Adjusting the temperature stands as a common practice for introducing variability in LLM responses (Miotto et al., 2022). However, consensus is lacking on the optimal temperature setting in simulating psychological characteristics. Existing research has explored various temperature settings, with common values including 0.7 (Garcia et al., 2024; Zhao et al., 2024) and 0 (Leng et al., 2024; Ren et al., 2024; Zhao et al., 2024). Some researchers advocate for higher

temperatures to boost creativity (Salewski et al., 2024), yet this can also introduce more noise into the data (Gunel et al., 2020). Conversely, setting the temperature to zero minimizes variability and enhancing replicability (Li et al., 2023), albeit restricting the ability to apply statistical analysis for investigation (Hagendorff et al., 2023). In our work, we examine the effect of two different temperature settings on model outputs.

**Evaluating the Quality of Persona Generation in LLMs:** Recent research has demonstrated the sophisticated ability of LLMs to generate and portray human-like personas (Binz and Schulz, 2023; Ouyang et al., 2022). These models can express psychological traits (Li et al., 2023; Stevenson et al., 2022) and simulate diverse populations (Deshpande et al., 2023; Salewski et al., 2024). However, evaluating the quality of these generated personas—particularly their coherence and psychological fidelity—remains challenging (Aher et al., 2023; Kovač et al., 2023). A key question, as noted by Ma et al. (2024), is whether the traits and attributes expressed by an LLM within a single session constitute a plausible, psychologically consistent persona. Researchers have developed several approaches to assess persona quality. Wang et al. (2024) conducts psychological interviews with LLM-generated personas to evaluate personality fidelity, while Gupta et al. (2024) employs a "judge" LLM to assess personas generated by other models. Another approach examines the consistency between LLM responses and their initial persona descriptions (Jiang et al., 2023). While these approaches focus on direct assessment of personas, our work takes a different approach by leveraging established patterns in human psychology, specifically the structure and interrelations of human values. By analyzing value correlations and their alignment with human data, we provide a quantitative framework for evaluating the psychological realism and internal consistency of LLM personas.

## 3 METHOD

In this section, we introduce the experimental design, models and prompts. The code and data are provided as supplementary files in the submission.

**The Value Questionnaire:** Our key goal was to assess the values interrelations of LLMs, similarly to how these are measured in humans. To do so, we applied the most estabilshed method for value estimation in human research: a values questionnaire. We applied the well validated and commonly used 57-item Portrait Value Questionnaire—Revised (PVQ-RR; Schwartz 2017), developed to measure the 19 values in the Schwartz's theory. The questionnaire describes fictional individuals and what is important to them. For example: "It is important to him/her to take care of people he/she is close to" (an item measuring benevolence-care values). For each such item, the subject is requested to indicate on a 6-point scale to what degree the persona they form is similar to the person described. Answers are categorical and range from a value of 1 (indicating "not like me at all") to 6 (indicating "very much like me"). See Appendix D for instructions and more example items from the questionnaire.

**Models Used:** We employed six prominent closed and open source LLMs: OpenAI's GPT-4-0314, Google's Gemini 1.0 Pro, Llama 3.1 8B, Llama 3.1 70B, Gemma 2 9B, and Gemma 2 27B. Each model was presented with each of our five prompt variants (see Section 3.1) 300 times, for a total of 1,500 runs per model. The prompts included gender-specific versions, with appropriate variants assigned based on the experimental condition. We conducted these experiments under two separate conditions: once with the temperature parameter set to 0.0 and once with it set to 0.7.

### 3.1 PROMPTS

To assess LLMs using the PVQ-RR, we first standardized the prompting approach after initial testing showed that minor wording variations produced consistent patterns in model responses. We adapted the standard PVQ instructions, adding explicit directives for the LLM to provide only value scores without elaboration, streamlining the data processing and analysis. The models were prompted to rate their similarity to each of the 57 PVQ-RR descriptions, which were presented together in a single administration. We also explored sequential prompting with the Llama models, presenting items individually, with no significant differences observed in our statistical analyses (see Appendix I, Figure 6 and Table 5).

We study a basic generic prompt, as well as four other prompts aimed at creating varied multiple personas:

**Basic prompt:** This prompt mirrors the adapted instructions of the PVQ-RR questionnaire without additional modifications. The prompt is structured as follows: "*For each of the following descriptions, please answer how much the person described is like you from 1 (Not like me at all) to 6 (Very much like me), without elaborating on your reasoning.*"

**Value Anchor prompt:** This prompt adds an anchor of value importance using identification with an item used in an additional value questionnaire, akin to the approach outlined in the study by Jiang et al. (2023). Participants are instructed as follows: "*For each of the following descriptions, please answer how much the person described is like you from 1 (Not like me at all) to 6 (Very much like me), without elaborating on your reasoning. Answer as a person that is [value]*". Here "[value]" is taken from the Best-Worst Refined Values scale (Lee et al., 2019). As a result, the prompts refer conceptually to the same values that are measured using the PVQ-RR, yet do not refer directly to the value items to be answered in response to the prompt. Examples of these anchor items include "protecting the natural environment from destruction or pollution" (universalism-nature) or "obeying all rules and laws" (conformity-rules). Please refer to Appendix E for the complete list of anchor items.

**Demographic prompt:** To investigate the influence of demographic context on LLM value consistency, we incorporate demographic details into the prompts, drawing upon established methodologies in persona-based LLM evaluation. Consistent with Argyle et al. (2023), this prompt extends the basic prompt by incorporating additional demographic details: "*For each of the following descriptions, please rate how much the person described is like you, using a scale from 1 (Not like me at all) to 6 (Very much like me), without elaborating on your reasoning. Answer as a [age]-year-old who identifies as [gender], working in the field of [occupation], and enjoys [hobby].*" The age, gender, occupation and hobby were randomly allocated for each prompt from a predefined list or range. The age range specified was between 18 and 75, with gender options including male, female, non-binary, and other, adapted from the National Academies of Sciences, Engineering, and Medicine (National Academies of Sciences, Engineering, and Medicine, 2022). Occupations were sourced from the World Values Survey (WVS-7; Haerpfer et al. 2022), while hobbies were chosen from established lists supplied by The Activity Card Sort (ACS-UK; Laver-Fawcett et al. 2016). The lists of occupations and hobbies are presented in Appendix F.1 and F.2.

**Generated Persona prompt:** In line with the methodology of Cheng et al. (2023), we directed the models to craft personas. Our instruction was formulated as: "*Create a persona (2-3 sentences long):*", with the temperature set at $0.7$ to increase creativity. An example of a persona generated by Gemini 1.0 Pro is as follows: "Emily is a 25-year-old marketing manager who is passionate about her career and loves spending time with her friends and family. She is always looking for new ways to improve her skills and knowledge, and she is always up for a challenge." Using these generated personas, we subsequently prompted the model as follows: "*For each of the following descriptions, please rate how much the person described is like you, using a scale from 1 (Not like me at all) to 6 (Very much like me), without elaborating on your reasoning. Answer as: [persona].*"

**Names prompt:** In line with a study by Aher et al. (2023), the prompts included titles (i.e., Mr., Ms., and Mx.) followed by surnames representing five distinct ethnic groups. From the 500 names cataloged in the previous study, we randomly generated 300 unique combinations of titles and names, including 60 from each ethnic group. The prompt was structured as follows: "*For each of the following descriptions, please rate how much the person described is like you, using a scale from 1 (Not like me at all) to 6 (Very much like me), without elaborating on your reasoning. Answer as [title + name]*". The complete list of titles and names is presented in Appendix F.3.

## 3.2 DATA ANALYSIS

In what follows we use the following notation. Let $V = 19$ be the set of value types studied. Each question in the questionnaire pertains to a particular item within the set of values $i \in V$. Furthermore, for each value there are $R = 3$ question variants. See Section B in the Appendix for example variants. Recall that the answer to each question is a number on a 6-point scale. For each LLM and prompt type, we presented the questionnaire $N$ times. The difference between each of these could be different personas, names, temperature sampling etc. Thus the overall set of answers for each LLM corresponds to a set of values $X_{i,j,k} \in \{1, \ldots, 6\}$ where $i = 1, \ldots, V$, $j = 1, \ldots, R$ and $k = 1, \ldots, N$.

When comparing to human data, we used the study in Schwartz and Cieciuch (2022). The data is from the study of 49 cultural groups.[2] The total number of participants was 53,472, the mean age was 34.2, ($SD = 15.8$), with 59% females. This dataset is publicly available through the Open Science Framework as described in Schwartz and Cieciuch (2022).

### 3.2.1 VALUE RANKINGS

Our first question for analysis was whether universal value hierarchy (Schwartz and Bardi, 2001; Schwartz and Cieciuch, 2022) is reflected in LLM resonses. Namely, do LLMs tend to rank the values more or less as human subjects do.

To obtain LLM rankings for a given set of LLM answers, we assigned a score $v_i$ to value $i$, where $v_i$ was the average score given to the three items measuring this value by the LLM (i.e. the average of $X_{i,\cdot,\cdot}$). From this score, we subtracted the average score given to all value items within the conversation, thus centering the data. We note that centering is the recommended practice in value research (Schwartz, 1992; Sagiv and Schwartz, 2022), and allows comparison to human samples. We then use these $v_i$ values to rank values. Finally, we calculated the Spearman's Rank Correlation ($\rho$) between this and the known human ranking (Schwartz and Cieciuch, 2022). We note that this analysis does not consider correlations between answers given in the same session, and thus it may be viewed as analyzing the first-order statistics of the responses.

### 3.2.2 CORRELATIONS BETWEEN VALUES

A key focus of our work is the correlation structure between values. That is, the question of whether the choice of value $i$ is correlated with that of value $j$. In humans, there is a robust correlation structure in which certain values are more strongly correlated than others. A key method for analyzing and visualizing correlation structures in value research is Multidimensional Scaling (MDS; Borg et al. 2018). MDS is a well-established and robust statistical technique that allows us to represent the complex interrelations between values in a lower-dimensional space, facilitating comparison and interpretation.

MDS analysis is performed as follows. First, the matrix $C \in R^{19 \times 19}$ of empirical correlation coefficients is formed. Next, each of the values is embedded into $\mathbb{R}^2$ via MDS, such that the distances in $\mathbb{R}^2$ best approximate the correlations. For human data, this results in an approximately circular embedding, as shown in Schwartz and Cieciuch (2022); Skimina et al. (2021b); Daniel and Benish-Weisman (2019). Here, we performed this analysis on the LLM data. To compare the resulting dataset to the human samples, we need to normalize for the degrees of freedom of rotation and translation. This is done via Procrustes Analysis between the human and LLM embeddings. The resulting embeddings were plotted. Then, we computed the sum of squared differences between the procrusted MDS locations of each value to the human benchmark. Larger differences indicate stronger divergence from the human samples.

## 4 RESULTS

The above analyses were performed for all models and prompting strategies. We checked that model responses only contained scores for the questions in the questionnaires, and that they could therefore be transformed to tabular form and analyzed. This was almost always the case except for Gemma 2 27B on the Demographic prompt at temperature 0.0, and we therefore do not provide results for that settings.

**Value Rankings:** As previously mentioned, research across different samples and cultures have shown that while individual differences exist in human value priorities, there are also robust common patterns. In this section, we analyze the LLM responses and compare them to the typical ranking of human values, as discussed in Section 3.2.1.

---

[2]Although the values benchmark was not drawn from a representative sample, it includes the most comprehensive dataset of values available. We validated our benchmark's representativeness by comparing its value rankings with those obtained from representative samples included in the European Social Survey (ESS, 2024), finding a strong correlation ($\rho = 0.79, p = 0.006$) that suggests robust value hierarchies across demographics.

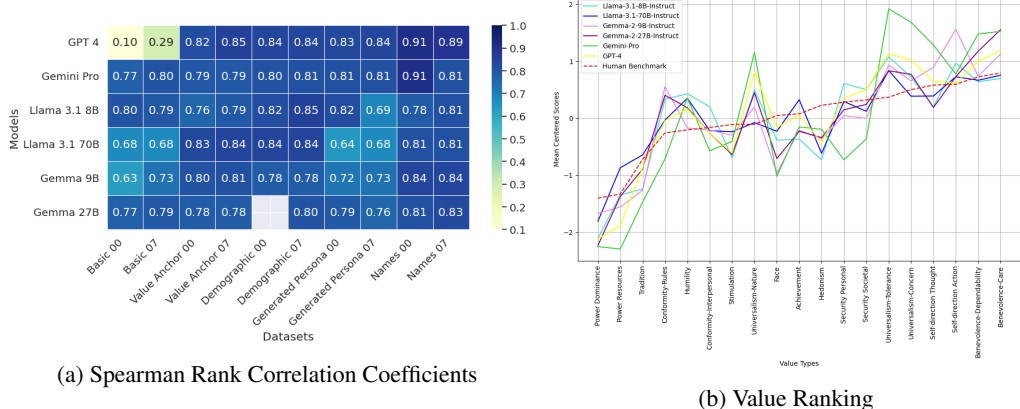

(a) Spearman Rank Correlation Coefficients

(b) Value Ranking

Figure 2: **Left:** A heatmap of Spearman rank correlation between human benchmark value hierarchies and dataset rankings for GPT-4-0314, Gemini 1.0 Pro, Llama 3.1 8B and 70B instruct, and Gemma 2 9B and 27B under two temperature conditions (0.0 and 0.7). **Right:** Average value scores for the Value Anchoring prompt at zero temperature. The x-axis shows values ordered according to human ranking (i.e., Power ranks lowest for humans and Benevolence ranks highest). The y-axis is the mean-centered scores the models ascribe to these values in the questionnaire, and human values in red. It can be seen that models tend to give lower scores to values that are ranked lower by humans, and higher scores to values ranked higher. The LLM scores also track the human scores (red curve) quite well.

Figure 2a shows the Spearman rank correlations between human rankings and those of the different models and prompting schemes. The results show high correlation levels ($> 0.8$) for many prompt-model combinations, with strong statistical significance ($p < .001$) across most models and prompts. This was particularly pronounced for the Value Anchor prompt, where correlations ranged from $0.75$ to $0.85$. One exception is the basic prompt with the GPT model, which shows very low correlation. Full rankings are provided in Appendix G for several models and prompts (see Table 2, Table 3 and Table 4). These reveal that values such as Benevolence that are highly ranked in humans are indeed also highly ranked by most LLMs (e.g,. ranked third and first by GPT-4-0314 for the Value Anchor prompt with temperatures $0.0$ and $0.7$ respectively). Conversely, values such as Power Dominance that are ranked low by humans, are ranked low by models (e.g., 19 by GPT-4-0314 for the Value Anchor prompt). Figure 2b shows the scores generated with the Value Anchor prompt, when sorted according to human preferences. It can be seen that the models tend to agree with the human ordering on the low and high ranked values. Taken together, these results demonstrate that LLMs tend on average to align with the human ranking of values.

**Correlations Between Values**   The MDS analysis (see Section 3.2.2) maps all values into $\mathbb{R}^2$ in a way that reflects their correlations. Here, we conduct MDS analyses for both human responses and LLM output, and then compare the results. The analyses were performed separately for each prompt, temperature, and model.

Looking at Figure 3, it can be seen that among humans, the values are organized in a circle in the theoretically expected order. These results have been consistently identified over the years and interpreted as resulting from individuals' aspiration to maintain personal consistency in their motivations (Schwartz, 1992). The figure compares human MDS configurations with those from Gemini 1.0 Pro at temperature $0.0$ using both the Value Anchor and Names prompts. It is evident that the MDS configuration resulting from the Value Anchor prompt more closely follows the human circular pattern than the MDS resulting from the Names prompt.

Our quantitative analyses across all models confirm this visual observation. The Value Anchor prompt showed particularly strong correlations with human value structures ($r = 0.87 - 0.95, p < .001$), providing robust statistical support for the similarity between LLM and human value patterns. To further quantify these comparisons, we calculated the mean squared difference between each pair of human and prompting method MDS matrices (i.e., matrices in $\mathbb{R}^{19 \times 2}$). Specifically, for each prompting method, we computed the squared Euclidean distance between the two-dimensional

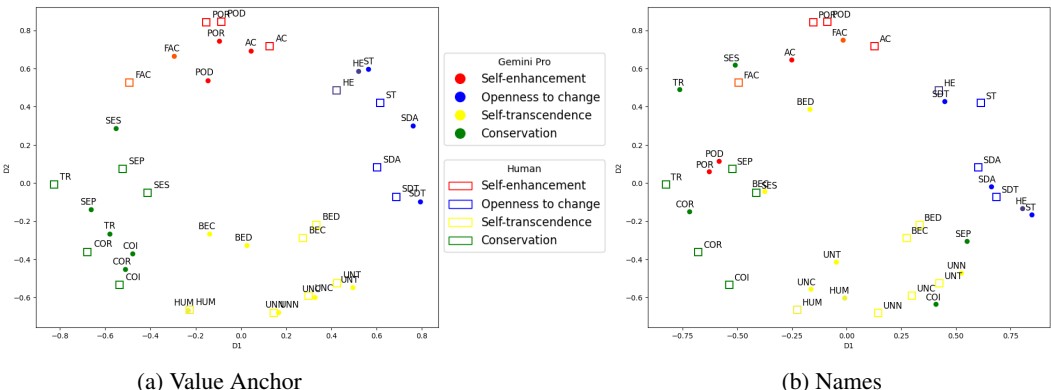

(a) Value Anchor                                          (b) Names

Figure 3: Figure 3: Comparison of Procrustes Analysis results, *visualizing value correlation structures* between human data (Schwartz and Cieciuch, 2022) and Gemini 1.0 Pro for Value Anchor and Names prompts, at temperature 0.0. The sum of squared differences, which measures the fit to human data, is 0.11 for the Value Anchor and 0.71 for the Names, indicating a better fit for the Value Anchor's correlation structure to human data. For acronyms of the values, refer to Appendix C.

|  | Basic | Value Anchor | Demographic | Persona | Names |
|---|---|---|---|---|---|
| GPT-4-0314 |  |  |  |  |  |
| 00 | 0.92 | **0.23** | 0.53 | 0.25 | 0.32 |
| 07 | 0.88 | **0.22** | 0.74 | **0.22** | 0.28 |
| Gemini 1.0 Pro |  |  |  |  |  |
| 00 | 0.87 | **0.11** | 0.42 | 0.39 | 0.71 |
| 07 | 0.69 | **0.11** | 0.75 | 0.28 | 0.57 |
| Llama 3.1 8B |  |  |  |  |  |
| 00 | 0.80 | **0.18** | 0.47 | 0.58 | 0.60 |
| 07 | 0.57 | **0.16** | 0.47 | 0.58 | 0.57 |
| Llama 3.1 70B |  |  |  |  |  |
| 00 | 0.61 | **0.10** | 0.29 | 0.37 | 0.45 |
| 07 | 0.44 | **0.10** | 0.22 | 0.40 | 0.44 |
| Gemma 2 9B |  |  |  |  |  |
| 00 | 0.42 | **0.10** | 0.19 | 0.39 | 0.23 |
| 07 | 0.82 | **0.11** | 0.16 | 0.32 | 0.12 |
| Gemma 2 27B |  |  |  |  |  |
| 00 | NA | **0.16** | NA | 0.31 | 0.23 |
| 07 | 0.64 | 0.17 | **0.15** | 0.25 | 0.19 |

Table 1: Sum of squared difference between the MDS embeddings of humans and LLM value measurements. Gemma 2 27B did not produce parseable results for the Demographic prompt, and for Gemma 27B at temperature 0.0, some values had zero-variance, thus precluding computation of correlation coefficients. All Llama models are Instruct.

coordinates of the human MDS solution and the corresponding LLM MDS solution. These results are presented in Table 1. A notable trend emerges: the Value Anchor prompt consistently produces a significantly lower sum of squared differences – indicating a closer alignment with human value correlations. Thus, emphasizing a single value leads to a demonstrably more human-like correlation structure than other prompting methods. Results for all other MDS plots for all prompts are included in Appendix H.

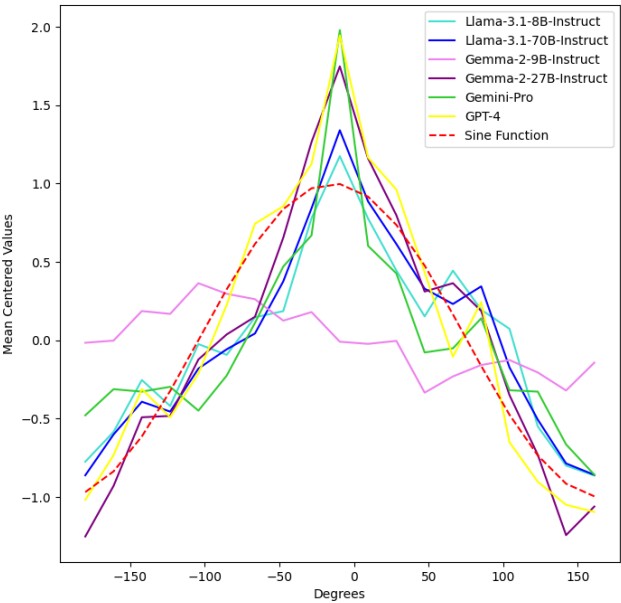

Figure 4: Analysis of scores after value anchoring. The plot shows the average of the score values after shifting to the anchored value. It can be seen that the anchored value receives the highest score, as expected. More surprisingly, neighboring values receive similarly high values, whereas more distant values receive lower values.

## 4.1 UNDERSTANDING VALUE ANCHORING

To better understand how value anchoring affects value correlations we performed additional analysis. Our analysis below reveals that value anchoring improves the alignment between model and human value correlation structures by systematically influencing how models score related values. When anchored to a specific value, models not only assign it a higher score but also manifest other values based on their proximity to the anchor in the theorized value circle (Figure 1). That is, values closer to the anchor receive higher scores, while distant values receive lower scores, thereby strengthening correlations between conceptually related values.

To quantify this relationship, we analyzed the scoring patterns by arranging the 19 anchoring values according to their circular order (Figure 1). For each Value Anchor prompt response, we normalized the scores by setting the anchored value to zero and computed the mean of these normalized scoring patterns across all 19 anchor conditions. The resulting patterns, shown in Figure 4, follow a sinusoidal function across all models except Gemma-2-9B. This analysis confirms that scores decrease systematically with increasing circular distance from the anchor value, demonstrating why Value Anchoring successfully captures human value correlation patterns.

## 5 DISCUSSION

Our study investigated how Large Language Models (LLMs) express human values through ranking and correlation analyses, highlighting the crucial role of prompt engineering in generating value profiles that align with human patterns. When presented with the PVQ-RR questionnaire without contextual framing (i.e., the Basic prompt condition), the models showed minimal variance across generated personas and demonstrated inconsistent responses to items measuring identical values. This was particularly evident in GPT-4-0314's responses, which showed stark deviations from human value hierarchies—notably prioritizing Conformity-Rules (ranked first versus humans' 16th) and deprioritizing Benevolence-Care (ranked 18th versus humans' first). These findings suggest that without proper contextualization, LLMs generate responses that likely reflect training data artifacts rather than coherent value systems. Notably, we observed high consistency across different model types, including both commercial and open LLMs.

The introduction of personality-oriented prompts improved the consistency of value profiles to varying degrees. While the value hierarchy remained consistent across prompts, indicating LLMs can simulate population-level value rankings, we found more variability in inter-value correlations. The Values Anchor prompt proved most effective in maintaining consistency across values within individual sessions. This suggests that with appropriate prompting, LLMs can generate a diverse "population" of individuals, each expressing distinct yet coherent value priorities. Perhaps our most striking finding was the emergence of human-like value correlations that aligned with the Schwartz circular model. This alignment emerged organically rather than through explicit prompting, suggesting LLMs possess a deeper, implicit understanding of human value structures.

Importantly, none of our prompts explicitly instructed LLMs how to respond regarding all values. Even the Value Anchor prompt instructed the the LLM with regards to reference to one value only. This suggests that LLMs not only follow instructions but use them as contextual frameworks to guide consistent responses across various values. This raises an intriguing question about how LLMs develop such clear value profiles. These patterns may emerge during pre-training. Previous studies have identified values in texts like newspaper articles and social media (Bardi et al., 2008; Ponizovskiy et al., 2020; Kumar et al., 2018). However, these earlier findings didn't necessarily reflect the theoretical value interrelations we observed. Unlike texts that might present multiple sides of value dilemmas, individuals tend to resolve such conflicts over time, leading to more coherent value systems (Bardi et al., 2009; Daniel and Benish-Weisman, 2019). The context-aware processing of LLMs may enable more accurate identification of value interrelations as compared to traditional lexical approaches. Alternative explanations include learning during fine-tuning or RLHF (Qiu et al., 2022)—a distinction that warrants further investigation through training source analysis and checkpoint evaluation.

Building on previous research into LLM persona consistency (Wang et al., 2024), we demonstrate that the unique qualities and solid empirical foundation of human values make them ideal for assessing persona stability (Sagiv and Schwartz, 2022; Sagiv et al., 2017; Knafo-Noam et al., 2024). Our findings suggest that known behavioral correlations in humans can effectively evaluate LLM persona consistency. While we focused on questionnaire-based evaluation, this framework could extend to other personality features.

Our methodology offers valuable contributions to psychological research. Researchers can use Value Anchoring to generate large-scale datasets simulating diverse value orientations, enabling hypothesis testing, instrument refinement, and replication studies. For LLM development, our value-correlation framework provides a novel metric for assessing human-likeness in generated text, particularly for dialogue systems. Future research could explore replicating known findings (such as age-related value differences) or testing novel hypotheses about value-behavior associations. The inclusion of both commercial and open LLMs enhances reproducibility and creates opportunities for downstream research in psychology and social sciences.

The societal implications of values in LLMs warrant careful consideration. While our results show that LLMs generally reproduce international value rankings, even subtle variations in value importance can significantly impact social dynamics, from gender roles (Lomazzi and Seddig, 2020), to entrepreneurship (Woodside et al., 2020), prosocial behavior (Daniel et al., 2020), and antisocial behavior (Benish-Weisman, 2019). Future research should examine how these values influence both LLM responses and human-AI interactions.

Several limitations merit attention. Our study examined a finite set of contexts (five prompts, two temperatures, six models), and while we found consistent patterns, broader contextual investigation could further enhance output quality. Additionally, while value rankings may vary across populations, the fundamental structural relationships revealed through MDS analysis likely remain stable, aligning with Schwartz's theory (Sagiv and Schwartz, 2022). We must also consider how training data biases might affect value expression across demographic groups. Finally, our focus on the PVQ-RR questionnaire and classification-based value elicitation suggests opportunities for exploring other value measurement approaches and more open-ended generation settings.

In conclusion, our research demonstrates that appropriate prompting techniques, particularly Value Anchoring, enable LLMs to generate remarkably coherent and human-like value profiles. This finding opens new possibilities for psychological and social science research, providing valuable tools for simulating human value systems and generating data for further investigation.

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

Table 2: Comparative analysis of 19 values' relative importance of the Value Anchor and Names datasets across temperatures for GPT-4-0314 and Gemini 1.0 Pro.

| | Benchmark Human Data | | GPT-4-0314 | | | | | | | | Gemini 1.0 Pro | | | | | | | |
| | | | Value Anchor 00 | | Value Anchor 07 | | Names 00 | | Names 07 | | Value Anchor 00 | | Value Anchor 07 | | Names 00 | | Names 07 | |
| Rank | Values | Mean | Mean | Rank | Mean | Rank | Mean | Rank | Mean | Rank | Mean | Rank | Mean | Rank | Mean | Rank | Mean | Rank |
|---|---|---|---|---|---|---|---|---|---|---|---|---|---|---|---|---|---|---|
| 1 | BEC | 0.79 | 1.18 | 3 | 1.24 | 1 | 0.66 | 3 | 0.65 | 3.5 | 1.52 | 3 | 1.44 | 3 | 1.56 | 3 | -0.10 | 13 |
| 2 | BED | 0.72 | 0.92 | 5 | 1.00 | 4 | 0.66 | 3 | 0.65 | 6 | 1.39 | 4 | 1.30 | 4 | 1.48 | 5 | 0.03 | 12 |
| 3 | SDA | 0.60 | 0.59 | 7 | 0.55 | 7 | 0.66 | 3 | 0.65 | 3.5 | 0.88 | 7 | 0.79 | 7 | 1.20 | 7 | 0.93 | 1 |
| 4 | SDT | 0.58 | 0.70 | 6 | 0.69 | 6 | 0.66 | 3 | 0.65 | 1.5 | 1.33 | 5 | 1.17 | 5 | 1.48 | 4 | 0.47 | 5 |
| 5 | UNC | 0.50 | 1.12 | 3 | 1.07 | 3 | 0.66 | 3 | 0.65 | 1.5 | 1.68 | 2 | 1.63 | 2 | 1.68 | 1 | 0.74 | 3 |
| 6 | UNT | 0.37 | 1.26 | 1 | 1.20 | 2 | 0.66 | 6 | 0.65 | 5 | 1.93 | 1 | 1.86 | 1 | 1.63 | 2 | 0.45 | 6 |
| 7 | SES | 0.32 | 0.41 | 8 | 0.36 | 8 | 0.64 | 7 | 0.58 | 7 | -0.49 | 12 | -0.42 | 12 | 0.60 | 8 | 0.14 | 10 |
| 8 | SEP | 0.28 | 0.21 | 9 | 0.26 | 9 | 0.48 | 11 | 0.43 | 9 | -0.87 | 14 | -0.75 | 14 | -0.75 | 14 | 0.21 | 8 |
| 9 | HE | 0.23 | -0.31 | 14 | -0.33 | 14 | 0.50 | 9 | 0.41 | 10 | 0.04 | 9 | -0.02 | 9 | 0.04 | 10 | 0.57 | 4 |
| 10 | AC | 0.08 | 0.07 | 11 | 0.09 | 11 | 0.06 | 14 | 0.10 | 14 | -0.08 | 11 | -0.22 | 11 | -0.35 | 11 | -0.54 | 16 |
| 11 | FAC | 0.05 | -0.28 | 13 | -0.31 | 13 | 0.30 | 12 | 0.20 | 13 | -1.01 | 16 | -0.91 | 16 | -0.68 | 13 | 0.10 | 11 |
| 12 | UNN | -0.10 | 0.98 | 4 | 0.97 | 4 | 0.58 | 8 | 0.51 | 8 | 1.17 | 6 | 1.10 | 6 | 1.29 | 6 | 0.19 | 9 |
| 13 | ST | -0.11 | -0.48 | 16 | -0.44 | 16 | -0.10 | 15 | -0.14 | 15 | -0.07 | 10 | -0.09 | 10 | -0.68 | 12 | -1.03 | 18 |
| 14 | COI | -0.16 | -0.41 | 15 | -0.42 | 15 | -0.74 | 17 | -0.65 | 17 | -0.72 | 13 | -0.58 | 13 | -0.90 | 16 | 0.34 | 7 |
| 15 | HUM | -0.20 | 0.20 | 10 | 0.20 | 10 | 0.30 | 13 | 0.22 | 12 | 0.12 | 8 | 0.15 | 8 | 0.52 | 9 | 0.84 | 2 |
| 16 | COR | -0.26 | -0.21 | 12 | -0.18 | 12 | 0.48 | 10 | 0.41 | 11 | -0.95 | 15 | -0.79 | 15 | -1.18 | 17 | -0.33 | 14 |
| 17 | TR | -0.72 | -0.98 | 17 | -0.93 | 17 | -0.62 | 16 | -0.59 | 16 | -1.57 | 17 | -1.43 | 17 | -0.88 | 15 | -0.52 | 15 |
| 18 | POR | -1.33 | -1.91 | 18 | -1.83 | 18 | -2.60 | 18 | -2.58 | 18 | -2.19 | 19 | -2.12 | 19 | -3.09 | 19 | -0.83 | 17 |
| 19 | POD | -1.40 | -2.14 | 19 | -2.05 | 19 | -2.82 | 19 | -2.81 | 19 | -2.14 | 18 | -2.10 | 18 | -2.98 | 18 | -1.64 | 19 |

Table 3: Comparative analysis of values' relative importance for the Llama 3.1 8B and Llama 3.1 70B datasets across temperatures.

| | Benchmark | | Llama 3.1 8B | | | | | | | | Llama 3.1 70B | | | | | | | | |
| | Human Data | | Value Anchor 00 | | Value Anchor 07 | | Names 00 | | Names 07 | | Value Anchor 00 | | Value Anchor 07 | | Names 00 | | Names 07 | |
| Rank | Values | Mean | Mean | Rank | Mean | Rank | Mean | Rank | Mean | Rank | Mean | Rank | Mean | Rank | Mean | Rank | Mean | Rank |
|---|---|---|---|---|---|---|---|---|---|---|---|---|---|---|---|---|---|---|---|
| 1 | BEC | 0.79 | 0.74 | 4 | 0.87 | 3 | 0.62 | 8 | 0.82 | 4 | 0.80 | 2 | 0.80 | 2 | 0.87 | 3 | 0.83 | 5 |
| 2 | BED | 0.72 | 0.64 | 5 | 0.64 | 5 | 0.62 | 9 | 0.65 | 7 | 0.64 | 4 | 0.69 | 4 | 0.67 | 7 | 0.69 | 7 |
| 3 | SDA | 0.60 | 1.00 | 2 | 1.04 | 2 | 1.12 | 2 | 1.12 | 2 | 0.76 | 3 | 0.72 | 3 | 0.84 | 5 | 0.84 | 4 |
| 4 | SDT | 0.58 | 0.24 | 11 | 0.34 | 9 | 0.89 | 5 | 0.67 | 6 | 0.48 | 5 | 0.53 | 5 | 1.01 | 2 | 0.96 | 2 |
| 5 | UNC | 0.50 | 0.77 | 3 | 0.80 | 4 | 1.11 | 3 | 0.99 | 3 | 0.46 | 6 | 0.49 | 6 | 0.85 | 4 | 0.84 | 3 |
| 6 | UNT | 0.37 | 1.14 | 1 | 1.12 | 1 | 1.19 | 1 | 1.20 | 1 | 0.90 | 1 | 0.88 | 1 | 1.10 | 1 | 1.10 | 1 |
| 7 | SES | 0.32 | 0.34 | 9 | 0.47 | 8 | 0.91 | 4 | 0.81 | 5 | -0.01 | 11 | 0.01 | 10 | 0.05 | 11 | 0.08 | 11 |
| 8 | SEP | 0.28 | 0.62 | 6 | 0.58 | 6 | 0.84 | 6 | 0.64 | 8 | 0.23 | 9 | 0.23 | 9 | 0.34 | 8 | 0.39 | 8 |
| 9 | HE | 0.23 | -0.61 | 16 | -0.58 | 16 | -1.38 | 16 | -0.50 | 14 | -0.51 | 16 | -0.46 | 16 | -0.13 | 13 | -0.31 | 13 |
| 10 | AC | 0.08 | -0.37 | 13 | -0.36 | 13 | -0.27 | 13 | -1.20 | 12 | 0.32 | 8 | 0.26 | 8 | 0.23 | 10 | 0.24 | 9 |
| 11 | FAC | 0.05 | -0.46 | 14 | -0.58 | 15 | 0.75 | 15 | -0.63 | 16 | -0.24 | 14 | -0.35 | 14 | -0.60 | 15 | -0.66 | 15 |
| 12 | UNN | -0.10 | 0.55 | 7 | 0.53 | 7 | 0.56 | 11 | 0.48 | 9 | 0.01 | 10 | -0.03 | 11 | 0.29 | 9 | 0.15 | 10 |
| 13 | ST | -0.11 | -0.59 | 15 | -0.53 | 14 | -0.67 | 14 | -0.34 | 13 | -0.15 | 13 | -0.12 | 13 | -0.03 | 12 | -0.08 | 12 |
| 14 | COI | -0.16 | -0.00 | 12 | -0.14 | 12 | 0.61 | 10 | 0.29 | 10 | -0.30 | 15 | -0.37 | 15 | -0.89 | 17 | -0.74 | 16 |
| 15 | HUM | -0.20 | 0.44 | 8 | 0.29 | 10 | 0.69 | 7 | 0.47 | 10 | 0.35 | 7 | 0.29 | 7 | 0.74 | 6 | 0.81 | 6 |
| 16 | COR | -0.26 | 0.29 | 10 | 0.22 | 11 | 0.56 | 12 | 0.03 | 11 | -0.14 | 12 | -0.11 | 12 | -0.35 | 14 | -0.36 | 14 |
| 17 | TR | -0.72 | -1.28 | 17 | -1.28 | 17 | -2.01 | 18 | -1.53 | 17 | -0.68 | 17 | -0.66 | 17 | -0.82 | 16 | -0.82 | 17 |
| 18 | POR | -1.33 | -1.35 | 18 | -1.29 | 18 | -1.77 | 17 | -1.55 | 18 | -0.89 | 18 | -0.88 | 18 | -1.46 | 18 | -1.46 | 18 |
| 19 | POD | -1.40 | -2.12 | 19 | -2.11 | 19 | -2.88 | 19 | -2.56 | 19 | -1.86 | 19 | -1.81 | 19 | -2.39 | 19 | -2.36 | 19 |

Table 4: Comparative analysis of values' relative importance for the Gemma 2 9B and Gemma 2 27B datasets across temperatures.

| | | | | Gemma 2 9B | | | | | | | | | Gemma 2 27B | | | | | | | |
| | Benchmark | | | | | | | | | | | | | | | | | | | |
| | Human Data | | Value Anchor 00 | | Value Anchor 07 | | Names 00 | | Names 07 | | Value Anchor 00 | | Value Anchor 07 | | Names 00 | | Names 07 | |
| Rank | Values | Mean | Mean | Rank | Mean | Rank | Mean | Rank | Mean | Rank | Mean | Rank | Mean | Rank | Mean | Rank | Mean | Rank |
|---|---|---|---|---|---|---|---|---|---|---|---|---|---|---|---|---|---|---|
| 1 | BEC | 0.79 | 1.14 | 2 | 1.22 | 2 | 1.90 | 2 | 1.88 | 2 | 1.57 | 1 | 1.57 | 1 | 1.53 | 1 | 1.50 | 1 |
| 2 | BED | 0.72 | 0.73 | 5 | 0.78 | 5 | 1.29 | 4 | 1.24 | 4 | 1.16 | 2 | 1.17 | 2 | 1.29 | 3 | 1.25 | 3 |
| 3 | SDA | 0.60 | 1.55 | 1 | 1.58 | 1 | 1.77 | 3 | 1.77 | 3 | 0.84 | 4 | 0.84 | 4 | 1.50 | 2 | 1.46 | 2 |
| 4 | SDT | 0.58 | 0.90 | 4 | 0.89 | 3 | 1.93 | 1 | 1.91 | 1 | 0.30 | 8 | 0.27 | 8 | 1.20 | 4 | 1.18 | 4 |
| 5 | UNC | 0.50 | 0.67 | 6 | 0.65 | 6 | 1.03 | 6 | 1.05 | 5 | 0.83 | 5 | 0.81 | 5 | 1.02 | 6 | 1.04 | 5 |
| 6 | UNT | 0.37 | 0.94 | 3 | 0.84 | 4 | 1.03 | 5 | 1.02 | 6 | 0.92 | 3 | 0.89 | 3 | 1.02 | 5 | 1.02 | 6 |
| 7 | SES | 0.32 | -0.01 | 10 | 0.00 | 10 | 0.75 | 8 | 0.76 | 8 | 0.11 | 10 | 0.12 | 10 | 0.32 | 10 | 0.33 | 9 |
| 8 | SEP | 0.28 | 0.08 | 9 | 0.05 | 9 | 0.15 | 9 | 0.12 | 9 | 0.07 | 11 | 0.07 | 11 | 0.11 | 11 | 0.07 | 11 |
| 9 | HE | 0.23 | -0.33 | 15 | -0.32 | 15 | -0.54 | 13 | -0.55 | 13 | -0.23 | 13 | -0.23 | 13 | -0.55 | 13 | -0.49 | 13 |
| 10 | AC | 0.08 | -0.22 | 13 | -0.22 | 13 | -0.50 | 12 | -0.50 | 12 | -0.19 | 12 | -0.21 | 12 | -0.34 | 12 | -0.32 | 12 |
| 11 | FAC | 0.05 | -1.01 | 16 | -1.05 | 16 | -1.45 | 17 | -1.45 | 17 | -0.84 | 16 | -0.81 | 16 | -1.24 | 17 | -1.24 | 17 |
| 12 | UNN | -0.10 | 0.21 | 8 | 0.20 | 8 | -0.03 | 10 | 0.01 | 10 | 0.56 | 6 | 0.45 | 6 | 0.55 | 7 | 0.61 | 7 |
| 13 | ST | -0.11 | -0.27 | 14 | -0.24 | 14 | -0.72 | 14 | -0.77 | 14 | -0.49 | 15 | -0.42 | 15 | -0.72 | 14 | -0.77 | 15 |
| 14 | COI | -0.16 | -0.21 | 12 | -0.20 | 12 | -0.94 | 15 | -0.93 | 15 | -0.33 | 14 | -0.31 | 14 | -1.00 | 16 | -1.01 | 16 |
| 15 | HUM | -0.20 | -0.14 | 11 | -0.18 | 11 | -0.40 | 11 | -0.41 | 11 | 0.13 | 9 | 0.13 | 9 | 0.47 | 8 | 0.47 | 8 |
| 16 | COR | -0.26 | 0.55 | 7 | 0.63 | 7 | 0.84 | 7 | 0.85 | 7 | 0.30 | 7 | 0.29 | 7 | 0.34 | 9 | 0.31 | 10 |
| 17 | TR | -0.72 | -1.23 | 17 | -1.24 | 17 | -0.99 | 16 | -0.96 | 16 | -0.99 | 17 | -0.96 | 17 | -0.80 | 15 | -0.76 | 14 |
| 18 | POR | -1.33 | -1.63 | 18 | -1.62 | 18 | -2.53 | 18 | -2.48 | 18 | -1.41 | 18 | -1.41 | 18 | -2.00 | 18 | -1.96 | 18 |
| 19 | POD | -1.40 | -1.72 | 19 | -1.77 | 19 | -2.59 | 19 | -2.57 | 19 | -2.30 | 19 | -2.26 | 19 | -2.70 | 19 | -2.69 | 19 |

## A    ADDITIONAL FILES

The Python and R code used to generate our prompt sets and analyses is available on the Open Review website.

## B    QUESTION VARIANTS

For each value, we use three different question variants. For example, the three question variants formulated to assess individuals' alignment with the value type "Power Dominance" (i.e., relating to asserting authority and control over others) are as follows:

- Question 6: "He desires recognition for his abilities and seeks admiration for his actions."
- Question 29: "He prefers taking charge of situations and making decisions."
- Question 41: "He actively seeks positions of power and influence, valuing control and authority over others."

## C    VALUE ACRONYMS

The figures in the paper use the following value acronyms: SDT = Self-Direction Thought; SDA = Self-Direction Action; ST = Stimulation; HE = Hedonism; AC = Achievement; POD = Power-Dominance; POR = Power-Resources; FAC = Face; SEP = Security-Personal; SES = Security-Societal; TR = Tradition; COR = Conformity-Rules; COI = Conformity-Interpersonal; HUM = Humility; UNN = Universalism-Nature; UNC = Universalism-Concern; UNT = Universalism-Tolerance; BEC = Benevolence-Caring; BED = Benevolence-Dependability

## D    EXAMPLE PORTRAIT VALUE QUESTIONNAIRE

Figure 5 provides an example for the Portrait Value Questionnaire that was used in our study.

| The value measured | Item | | How much like you is this person? | | | | | |
| --- | --- | --- | --- | --- | --- | --- | --- | --- |
| | Male Version | Female Version | 1 | 2 | 3 | 4 | 5 | 6 |
| Self-direciton Thought | It is important to him to form his views independently. | It is important to her to form her views independently. | | | | | ✓ | |
| Security Societal | It is important to him to have a strong state that can defend its citizens. | It is important to her to have a strong state that can defend its citizens. | | | ✓ | | | |
| Hedonism | It is important to him to have a good time. | It is important to her to have a good time. | | | ✓ | | | |
| Conformity-Interpersonal | It is important to him never to annoy anyone. | It is important to her never to annoy anyone. | | ✓ | | | | |

Figure 5: Portrait Value Questionnaire—Revised - example items. The instructions provided were: "Here we briefly describe some people. Please read each description and think about how much each person is or is not like you. Tick the box to the right that shows how much the person in the description is like you". Rankings correspond to the following descriptions: 1-Not like me at all, 2-Not like me, 3-A little like me, 4-Somewhat like me, 5-Like me, 6-Very much like me.

## E  THE COMPLETE ITEM LIST OF BEST-WORST REFINED VALUES (BWVR)

In our value anchoring approach, we used the description of values in Lee et al. (2019) to prompt the LLMs. The set of descriptions is provided below.

1. **Self-direction-thought**: developing your own original ideas and opinions
2. **Self-direction-action**: being free to act independently
3. **Stimulation**: having an exciting life; having all sorts of new experiences
4. **Hedonism**: taking advantage of every opportunity to enjoy life's pleasures
5. **Achievement**: being ambitious and successful
6. **Power-dominance**: having the power that money and possessions can bring
7. **Power-resources**: having the authority to get others to do what you want
8. **Face**: protecting your public image and avoiding being shamed
9. **Security-personal**: living and acting in ways that ensure that you are personally safe and secure
10. **Security-societal**: living in a safe and stable society
11. **Tradition**: following cultural family or religious practices
12. **Conformity-rules**: obeying all rules and laws
13. **Conformity-interpersonal**: making sure you never upset or annoy others
14. **Humility**: being humble and avoiding public recognition
15. **Benevolence-dependability**: being a completely dependable and trustworthy friend and family member
16. **Benevolence-caring**: helping and caring for the wellbeing of those who are close
17. **Universalism-concern**: caring and seeking justice for everyone especially the weak and vulnerable in society
18. **Universalism-nature**: protecting the natural environment from destruction or pollution
19. **Universalism-tolerance**: being open-minded and accepting of people and ideas, even when you disagree with them
20. **Animal welfare**: caring for the welfare of animals

## F  SUPPLEMENTARY LISTS FOR PROMPTS

### F.1  LIST OF HOBBIES (LAVER-FAWCETT ET AL., 2016)

- Shopping
- Driving
- Taking care of pets
- Managing financial matters
- Taking a rest
- Going to the hairdresser / barber
- Childcare / babysitting
- Preparing a hot drink
- Conducting personal care
- Conducting personal business
- Taking care of others
- Cleaning/ fixing things
- Talking on the telephone
- Creative writing / keeping a journal
- Knitting / needlecrafts
- Playing table games
- Going to watch a sports event
- Cooking / baking as a hobby
- Doing puzzles / crosswords
- Using a computer
- Taking photographs
- Reading a religious book
- Written communications
- Looking at photo albums / home videos
- Researching family / local history
- Reading a newspaper / magazine
- Watching nature
- Playing bingo

- Watching television
- Listening to the radio / music
- Relaxing / meditating
- Entering competitions
- Reading a book
- Flower arranging
- Going to the beach
- Dancing
- Swimming
- Playing a ball game
- Walking
- Hiking / rambling
- Exercising
- Riding a bicycle
- Going on holiday / travelling
- Attending a leisure / social group
- Going to gardens / parks
- Fishing
- Having a picnic / BBQ
- Spending time with family / friends
- Eating out
- Going to parties
- Going for drinks at pubs / social clubs
- Volunteer work
- Cultural visits
- Going to music / performing arts events
- Going to church / mosque / synagogue / temple / other
- Collecting (stamps, posters, figures)
- Drawing / painting
- Interior decorating

### F.2 LIST OF OCCUPATIONS (HAERPFER ET AL., 2022)

- Professional and technical expertise
- Higher administrative roles
- Clerical work
- Sales
- Service
- Skilled work
- Semi-skilled work
- Unskilled work
- Farm work
- Farm management or ownership
- No prior job experience

### F.3 LIST OF TITLES AND NAMES (AHER ET AL., 2023)

#### AMERICAN INDIAN GROUP

- Ms. Haskie
- Ms. Secatero
- Mr. Goseyun
- Ms. Manuelito
- Ms. Delgarito
- Ms. Roanhorse
- Mx. Chasinghawk
- Mx. Notah
- Mr. Gishie
- Mx. Secody
- Mx. Bitsuie
- Mr. Goldtooth
- Ms. Henio
- Mx. Yellowhair
- Mx. Chinana
- Ms. Kanuho
- Mr. Clah
- Mx. Smallcanyon
- Ms. Peshlakai
- Mx. Tabaha
- Mx. Clitso
- Ms. Begaye
- Mx. Altaha
- Mr. Littlelight
- Mr. Tsinnijinnie
- Mr. Cayaditto
- Mx. Apachito
- Mx. Todacheenie
- Mr. Wauneka
- Mr. Begay
- Ms. Keams
- Mr. Etsitty
- Mx. Laughing
- Mx. Cosay
- Ms. Ganadonegro
- Ms. Hosteen
- Mr. Todacheene
- Ms. Twobulls
- Ms. Yazzie
- Ms. Blackgoat
- Ms. Tapaha
- Mr. Whiteplume

- Mr. Nez
- Mr. Whitehat
- Mx. Tsinnie
- Mr. Shije
- Ms. Becenti
- Mx. Blueeyes

- Mx. Tsosie
- Mr. Atcitty
- Mx. Manygoats
- Mr. Altaha
- Mx. Goseyun
- Mx. Begay

- Mr. Henio
- Mx. Whiteplume
- Ms. Goseyun
- Mr. Begaye
- Mr. Bitsuie
- Ms. Laughing

ASIAN PACIFIC GROUP

- Mr. Ha
- Mr. Hu
- Mr. Lin
- Mr. Kim
- Mx. Chau
- Mr. Ngo
- Mr. Tran
- Mx. Zhou
- Ms. Oh
- Ms. Le
- Ms. Moua
- Mr. Shen
- Mx. Wang
- Ms. Chung
- Mx. Chu
- Ms. Cheung
- Mx. Li
- Mx. Ng
- Mr. Kang
- Mx. Ko

- Ms. Gupta
- Mx. Thai
- Mr. Jiang
- Mx. Chen
- Mr. Luu
- Mr. Cheng
- Mr. Yan
- Mr. Lai
- Ms. Sun
- Mx. Ho
- Ms. Lo
- Ms. Duong
- Ms. Song
- Mx. Thao
- Ms. Yang
- Mr. Shin
- Mx. Jain
- Ms. Yu
- Mx. Chiu
- Mx. Sharma

- Mr. Xiong
- Mr. Huang
- Mx. Pham
- Ms. Yoon
- Ms. Choi
- Ms. Liu
- Mx. Zhang
- Mx. Vu
- Ms. Zhao
- Mr. Tam
- Ms. Nguyen
- Mx. Trinh
- Mr. Nguyen
- Mr. Moua
- Mr. Yoon
- Ms. Chiu
- Mx. Lin
- Mx. Moua
- Mx. Shin
- Mx. Luu

BLACK GROUP

- Mx. Cisse
- Ms. Jeanpierre
- Ms. Wigfall
- Mr. Calixte
- Mr. Conteh
- Mr. Jeanbaptiste
- Mr. Mondesir
- Ms. Mwangi
- Mr. Jeanjacques
- Ms. Jama
- Mx. Straughter
- Ms. Smalls
- Mx. Fofana
- Mr. Koroma
- Mx. Abdullahi

- Ms. Kebede
- Ms. Prioleau
- Mr. Manigault
- Mr. Mekonnen
- Mr. Gadson
- Ms. Diop
- Ms. Grandberry
- Mr. Njoroge
- Mr. Jalloh
- Ms. Sesay
- Mx. Jeanfrancois
- Mx. Jeancharles
- Ms. Jeanlouis
- Ms. Louissaint
- Mx. Traore

- Mr. Osei
- Mr. Bekele
- Ms. Bekele
- Mr. Straughter
- Mr. Grandberry
- Mr. Jeanlouis
- Mx. Mekonnen
- Mx. Conteh
- Mx. Bekele
- Ms. Gadson
- Mx. Smalls
- Mr. Louissaint
- Mx. Njoroge
- Mr. Kebele
- Mr. Prioleau

- Mx. Mwangi
- Ms. Cisse
- Mx. Diop
- Ms. Calixte
- Mr. Jama

- Ms. Straughter
- Ms. Osei
- Mx. Jeanjacques
- Mr. Mwangi
- Mr. Jeanfrancois

- Mx. Sesay
- Ms. Abdullahi
- Mx. Grandberry
- Mx. Osei
- Mr. Smalls

## HISPANIC GROUP

- Ms. Rios
- Mx. Gomez
- Mx. Delgado
- Ms. Lopez
- Ms. Avila
- Mr. Herrera
- Mx. Salazar
- Mr. Morales
- Mr. Guerrero
- Ms. Munoz
- Mr. Rojas
- Mr. Carrillo
- Mx. Aguilar
- Mx. Moreno
- Mr. Chavez
- Mx. Diaz
- Mx. Jimenez
- Mr. Fuentes
- Ms. Espinoza
- Mx. Perez

- Ms. Nunez
- Mx. Rivas
- Mr. Alvarez
- Ms. Garcia
- Mr. Molina
- Mr. Vasquez
- Ms. Medina
- Ms. Salinas
- Mr. Pena
- Mx. Ortega
- Mx. Rivera
- Ms. Torres
- Ms. Rodriguez
- Ms. Castillo
- Ms. Mejia
- Mx. Vargas
- Ms. Gonzalez
- Ms. Vega
- Mr. Vazquez
- Ms. Calderon

- Mr. Ramirez
- Mr. Ruiz
- Mx. Padilla
- Ms. Flores
- Mr. Aguirre
- Mx. Contreras
- Ms. Soto
- Mr. Sanchez
- Ms. Marquez
- Mr. Serrano
- Mx. Marquez
- Ms. Traore
- Mr. Munoz
- Mr. Contreras
- Mr. Lopez
- Mr. Torres
- Mr. Rivas
- Ms. Fuentes
- Mx. Molina
- Ms. Aguirre

## WHITE GROUP

- Mr. Duffy
- Mx. Olsen
- Mr. Hoffman
- Mr. Oneill
- Mr. Brennan
- Mx. Foley
- Mr. Mccarthy
- Ms. Bauer
- Ms. Parsons
- Mx. Russo
- Ms. Peck
- Mx. Snyder
- Ms. Carlson
- Mr. Rasmussen
- Ms. Berg
- Mx. Meyer

- Mx. Donovan
- Mr. Hoover
- Ms. Schultz
- Ms. Hensley
- Ms. Krueger
- Ms. Friedman
- Mx. Brandt
- Mr. Boyle
- Mx. Reilly
- Ms. Moyer
- Ms. Johnston
- Ms. Conrad
- Mx. Christensen
- Mr. Stark
- Ms. Koch
- Mx. Kline

- Ms. Weiss
- Ms. Owen
- Mx. Weber
- Mr. Schaefer
- Mx. Mcmahon
- Mr. Roth
- Mr. Hartman
- Mx. Schmidt
- Mx. Flynn
- Mr. Case
- Mx. McMahon
- Ms. Mayer
- Ms. Hebert
- Mr. Kramer
- Ms. Huber
- Mx. Larson

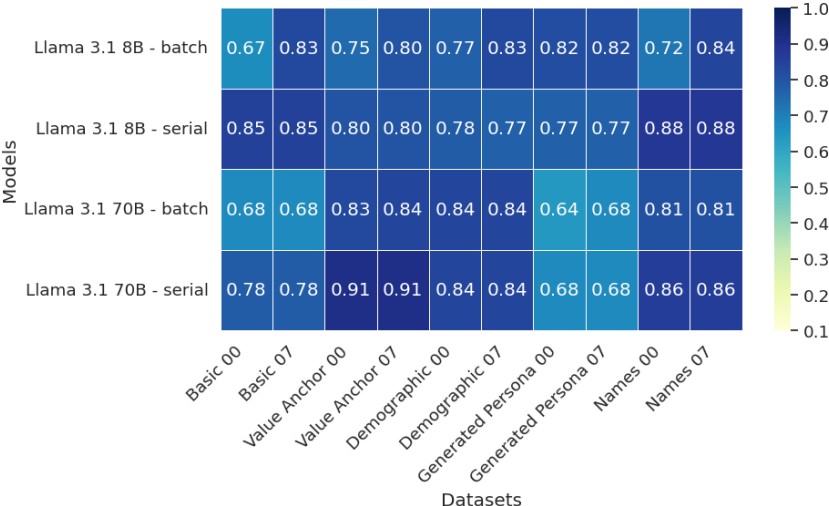

Figure 6: A heatmap of Spearman rank correlation between benchmark value hierarchies and dataset rankings for Llama 3.1 8B and 70B instruct for batch versus serial prompting methods, across temperature conditions.

- Mr. Nielsen
- Ms. Gallagher
- Mr. Howe
- Mr. Macdonald

- Mr. Morse
- Mr. Schneider
- Mr. Snyder
- Mr. Friedman

- Mx. Berg
- Ms. Hoover
- Mx. Case
- Ms. Schaefer

## G ADDITIONAL RESULTS FOR VALUE RANKINGS

Table 2, Table 3 and Table 4 provide additional results on value rankings for several prompting approaches.

## H ADDITIONAL MDS PLOTS

In the main text we provided the MDS plots for Gemini 1.0 Pro for Value Anchor and Names. Here we provide further plots for Gemini 1.0 Pro in Figure 7, all the GPT-4-0314 plots in Figure 8, all of the Llama 3.1 8B plots in Figure 9, all of Llama 3.1 70B plots in Figure 10, all of Gemma 2 9B plots in Figure 11, and Gemma 2 27B plots in Figure 12 for temperature 0.0.

## I COMPARING BATCH AND SEQUENTIAL PROMPTING

In the main text, we focused exclusively on batch prompting, where all items from the questionnaire were presented in a single prompt. An alternative is to present the questions in sequence, and ask the model to answer a question as soon as it is presented. To investigate potential differences between batch and sequential prompting, we evaluated on Llama models (in commercial models, sequential prompting is more expensive than batch). The value-ranking results are summarized in Figure 6, and the value-correlation results in Table 5. Regarding the values rankings, Fisher's $Z$ transformation tests revealed no significant differences were observed between batch and sequential prompting methods for either Llama 3.1 70B or Llama 3.1 8B across all tested categories. For Llama 3.1 8B, the closest to significance was in the Names with temperature 0.0 ($z = -1.32$, $p = .185$), while for Llama 3.1 70B, the Value Anchor with temperature 0.0 showed the largest non-significant difference ($z = -0.96$, $p = .337$). This indicates that, overall, the ranking correlations are closely aligned, with no clear inclination toward either batch or sequential prompting as better replicating the human value hierarchy. As for the value-correlations, it can be seen that the sequential prompts replicate the finding that the Value Anchor prompt best captures the circular structure of human values. Interestingly, for Llama 3.1 8B, batch prompting appeared to yield superior results. However, for Llama 3.1 70B, this was not the case across most prompts, suggesting that batch prompting may not consistently perform better across different models.

| Llama 3.1 8B | | | | |
|---|---|---|---|---|
| | Value Anchor | Demographic | Generated Persona | Names |
| **Batch prompting** | | | | |
| 00 | 0.18 | 0.47 | 0.58 | 0.60 |
| 07 | 0.16 | 0.47 | 0.58 | 0.57 |
| **Serial prompting** | | | | |
| 00 | 0.18 | 0.54 | 0.65 | 0.61 |
| 07 | 0.18 | 0.54 | 0.65 | 0.37 |
| **Llama 3.1 70B** | | | | |
| | Value Anchor | Demographic | Generated Persona | Names |
| **Batch prompting** | | | | |
| 00 | 0.10 | 0.29 | 0.37 | 0.45 |
| 07 | 0.10 | 0.22 | 0.40 | 0.44 |
| **Serial prompting** | | | | |
| 00 | 0.14 | 0.26 | 0.20 | 0.48 |
| 07 | 0.14 | 0.26 | 0.20 | 0.69 |

Table 5: Sum of squared difference for MDS embeddings of humans and LLM.

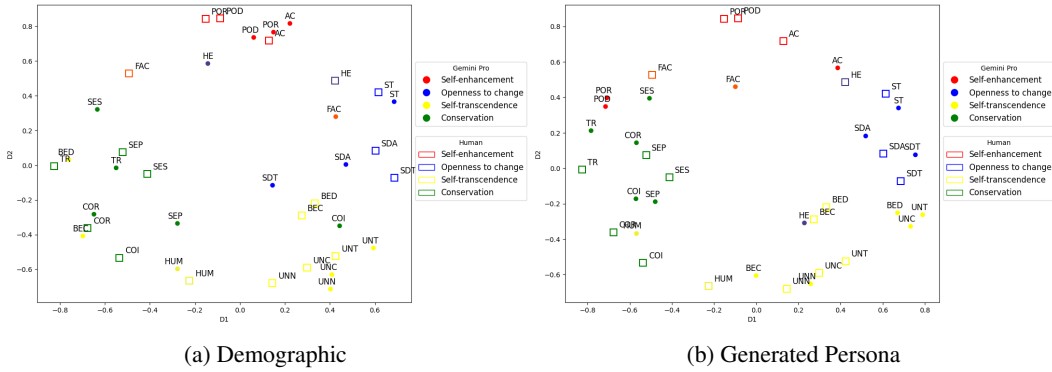

(a) Demographic  (b) Generated Persona

Figure 7: Comparison of the MDS results between human data (Schwartz and Cieciuch, 2022) and Gemini 1.0 Pro for Demographic and Generated Persona respectively, in the temperature 0.0 condition.

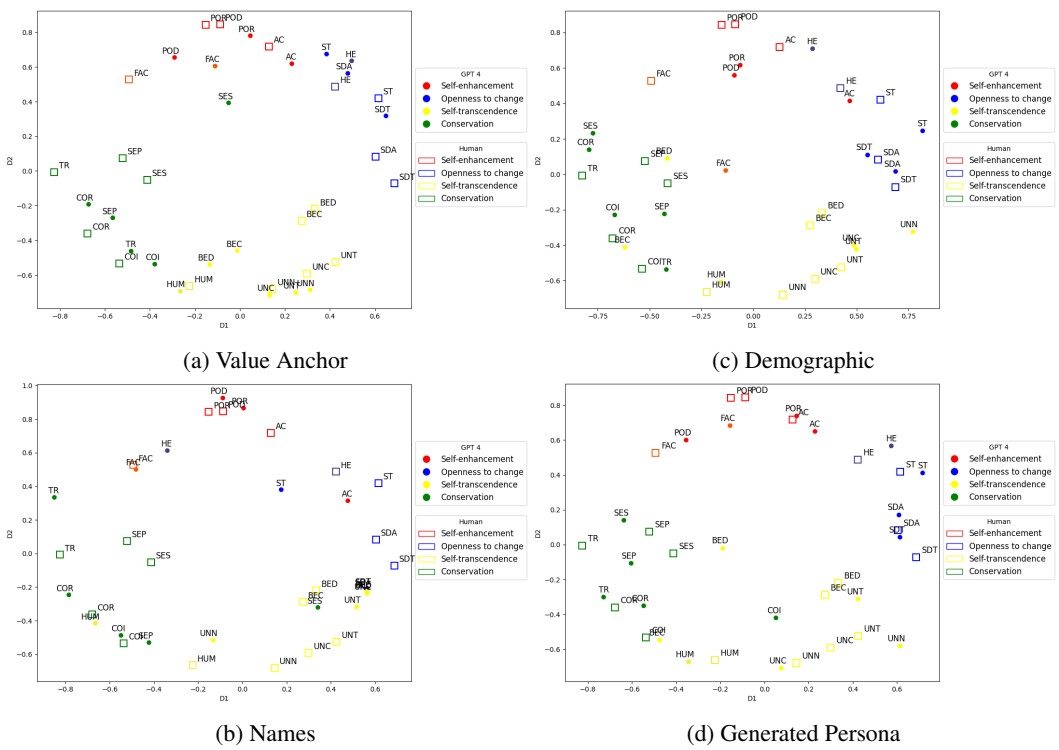

Figure 8: Comparison of the MDS results between human data (Schwartz and Cieciuch, 2022) and GPT-4-0314 for all prompts, in the temperature 0.0 condition.

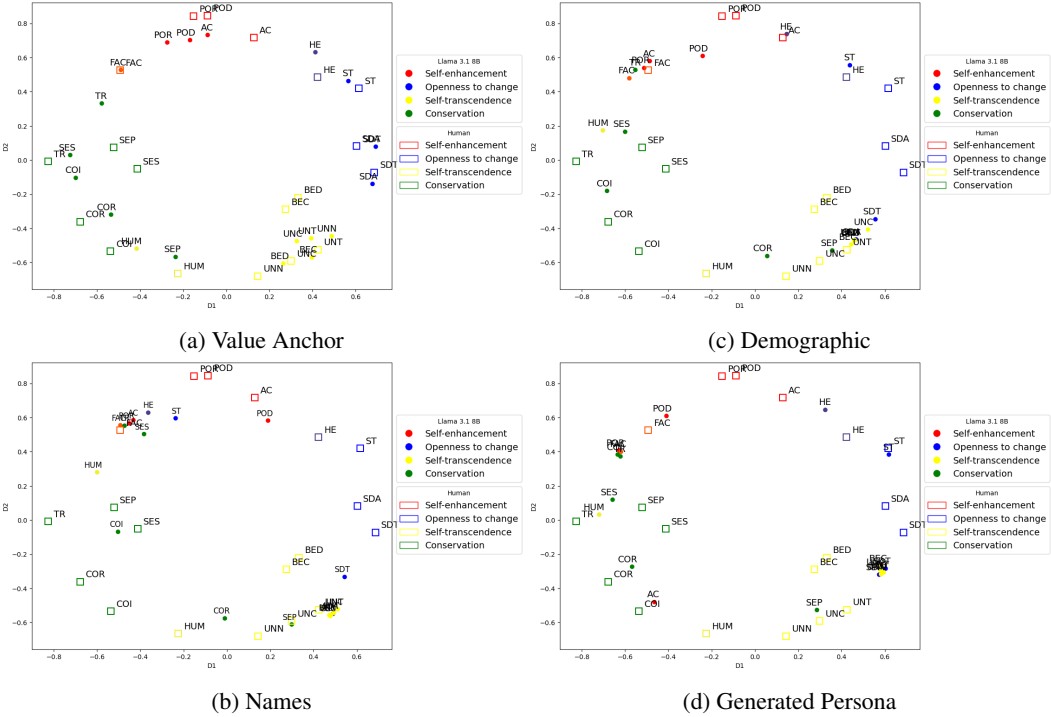

Figure 9: Comparison of the MDS results between human data (Schwartz and Cieciuch, 2022) and Llama 3.1 8B for all prompts, in the temperature 0.0 condition.

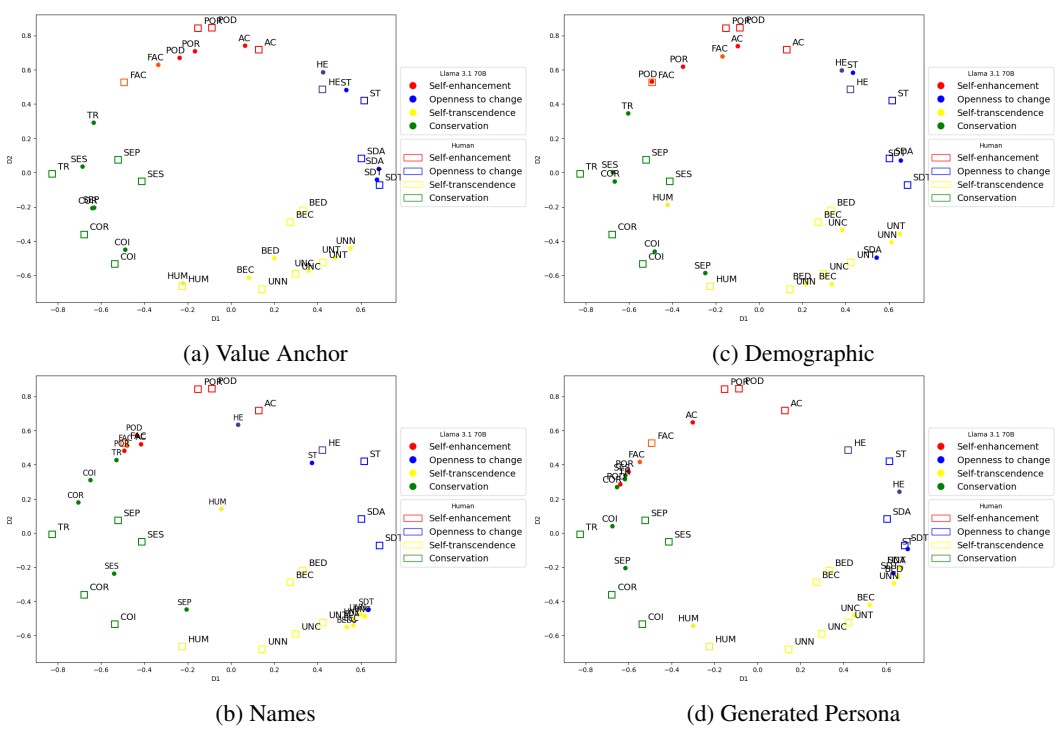

Figure 10: Comparison of the MDS results between human data (Schwartz and Cieciuch, 2022) and Llama 3.1 70B for all prompts, in the temperature 0.0 condition.

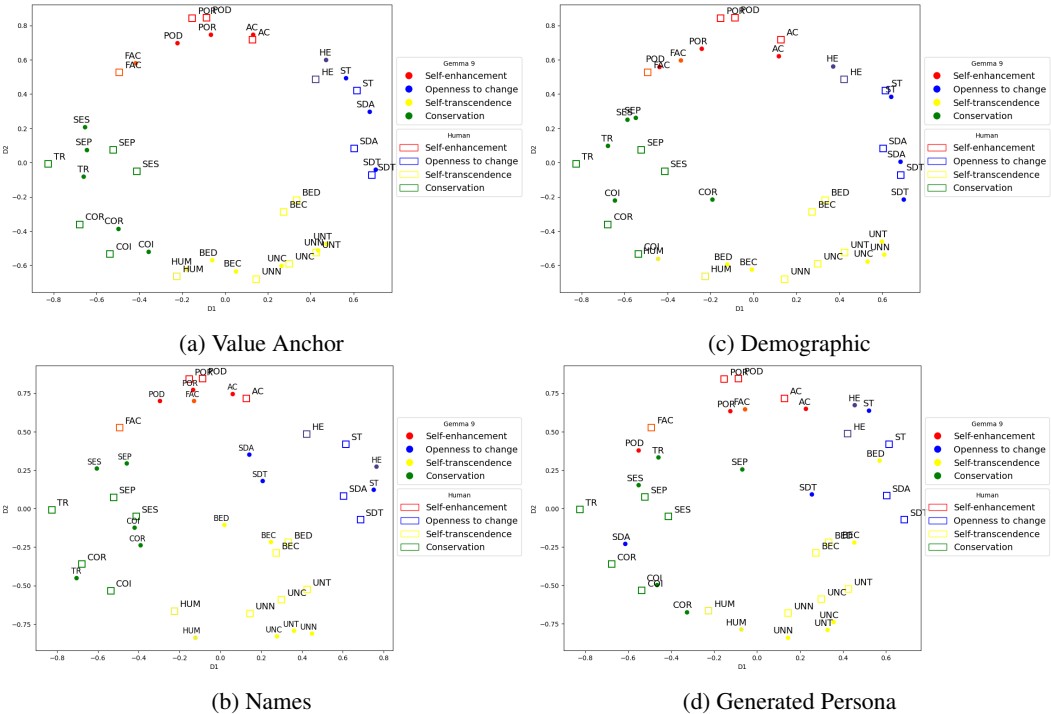

Figure 11: Comparison of the MDS results between human data (Schwartz and Cieciuch, 2022) and Gemma 2 9B for all prompts, in the temperature 0.0 condition.

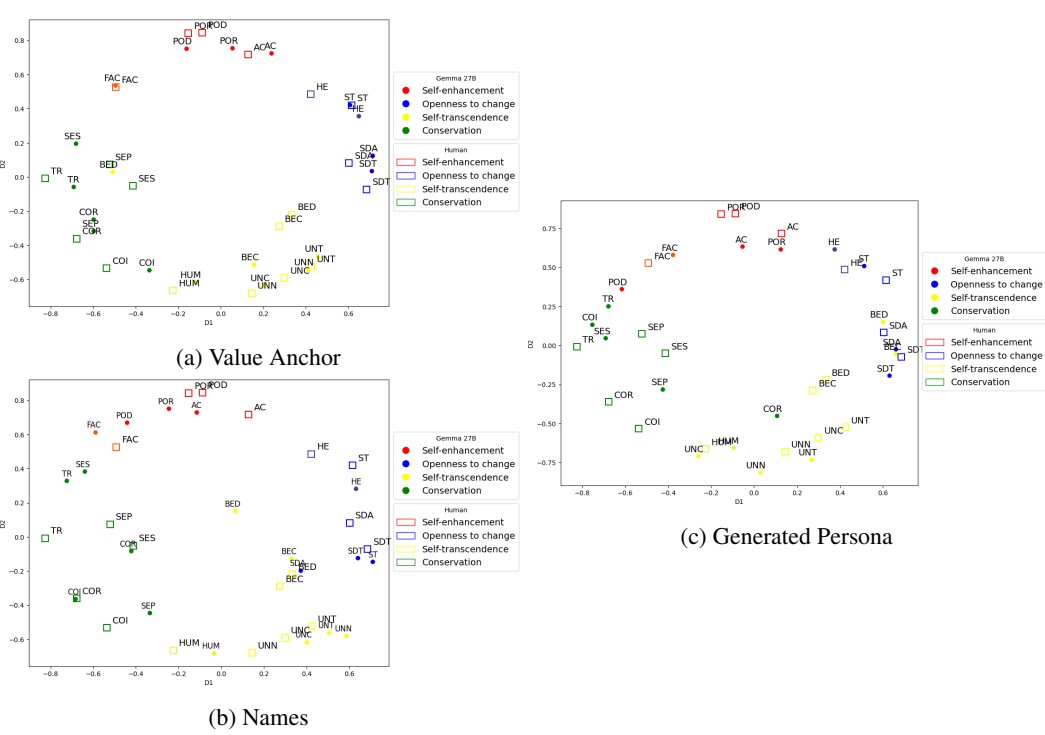

Figure 12: Comparison of the MDS results between human data (Schwartz and Cieciuch, 2022) and Gemma 2 27B for all prompts, in the temperature $0.0$ condition, with the exception of the Demographic prompt-(see Footnote 2).

