# OpenReview forum: "Do LLMs have Consistent Values?"
_ICLR.cc/2025/Conference — ICLR 2025 Poster_

### Official Review · Reviewer_uULc · 2024-10-19

**Soundness:** 2
**Presentation:** 2
**Contribution:** 2
**Rating:** 3
**Confidence:** 4

**Summary:**

- Investigates whether large language models (LLMs) exhibit consistent value structures similar to those found in humans, using well-established psychological frameworks.

- Uses the Portrait Value Questionnaire-Revised (PVQ-RR) to assess LLM responses, evaluating how closely they align with human value rankings and correlations. Six prominent LLMs, including GPT-4 and Gemini Pro, were tested under various prompting strategies (e.g., basic, Value Anchor, demographic).

- LLM responses, when prompted correctly (especially with the "Value Anchor" prompt), show high consistency with human value hierarchies, including correlations between values.

- The study reveals that LLMs can simulate value-driven personas and produce human-like value profiles with the right prompting, mirroring both first-order (value ranking) and second-order (value correlation) statistics.

- Mimic (?) coherent psychological profiles based on value systems, providing a novel method for assessing LLM consistency. The study also suggests broader implications for applying psychological theory to evaluate LLM behavior.

**Strengths:**

- While the discussion around value systems in LLMs is not entirely new, the paper builds on existing work by introducing the novel use of the Value Anchor prompt to assess how closely LLM responses align with human values. Unlike earlier research that uses game-based approaches (Kovač-2024), psychological surveys (Wang-2024), or repeated role-playing (Lee-2024), this paper systematically examines value rankings and correlations using well-established psychological frameworks. (Scherrer-2023) looks similar as it also deals with prompts, but (Scherrer-2023)  only deals with prompt format without a separate discussion of value anchoring. This paper's focus on both first-order (value ranking) and second-order (value correlations) statistics to assess consistency adds depth and precision to the existing discussion.

- I don't have specific complaints about the methodology. It seems to be a fairly well-executed set of experiments compared to other papers on human value systems. The use of the Portrait Value Questionnaire-Revised (PVQ-RR) ensures the analysis is grounded. Additionally, the introduction of comparative analysis across prompting strategies strengthens the study, as it demonstrates that the choice of prompt significantly affects LLM output and its coherence with human-like values.

(Kovač-2024) "Stick to your role! Stability of personal values expressed in large language models."

(Lee-2024) "Language Models Show Stable Value Orientations Across Diverse Role-Plays."

(Wang-2024) "Incharacter: Evaluating personality fidelity in role-playing agents through psychological interviews."

(Scherrer-2023) "Evaluating the Moral Beliefs Encoded in LLMs"

**Weaknesses:**

- The paper does not adequately address why it is a valuable addition to the already crowded discussion around values in LLMs. Several papers this year have explored similar themes of value consistency and expression through various methods (e.g., role-playing, moral beliefs, and novel-based approaches). The authors need to provide a clearer explanation of how their use of the Value Anchor prompt and focus on value correlations sets this study apart from others. It would strengthen the paper if the authors cited these related works more extensively and articulated how their approach advances the conversation rather than merely replicating it.

- One thing that I believe this paper fails to answer is that the paper leans heavily on the idea that LLMs should be compared to human value systems as the benchmark for consistency. However, it does not explore whether LLMs should necessarily be held to human standards, or whether they could develop a distinct and equally valid form of value coherence that differs from human psychology. By only focusing on human comparison, the paper misses an opportunity to explore how LLMs might create unique, non-human patterns of consistency that could still be valuable.

**Questions:**

na

---

### Official Review · Reviewer_8MQX · 2024-10-31

**Soundness:** 2
**Presentation:** 3
**Contribution:** 2
**Rating:** 6
**Confidence:** 4

**Summary:**

This paper dives into the intriguing question of whether LLMs can reflect human like values, both in how they rank certain values and how those values relate to one another. Using Schwartz’s Theory of Basic Human Values as a benchmark, the authors investigate how different ways of prompting, especially a “Value Anchor” technique, impact the models' responses. The results are promising - when given specific types of prompts, particularly the Value Anchor, LLMs tend to mirror human patterns of valuing and prioritizing. This suggests that with the right approach LLMs might be guided to exhibit more human-like consistency in values which could open up new opportunities for their use in applications where understanding of human values is key.

**Strengths:**

1. The paper is well-written and easy to follow.
2. The work is well-grounded in established psychological theory, particularly Schwartz's Theory of Basic Human Values.
3. The use of value rankings and correlations provides concrete, measurable ways to compare LLM outputs to human data.
4. The paper studies a timely and important issue.

**Weaknesses:**

1. The experimental results could be made stronger by analyzing whether minor variations in the same prompt could elicit the same results. Since language models often respond differently to small changes in wording, showing how the results hold up with different prompts would add a lot of value. A bit more discussion around this could help understand how stable the findings really are.

2. It would be great to see the value rankings and correlation structures explored in generation tasks as well, not just in classification. Since the goal here is to simulate different values and perspectives across populations, showing that language models can pick up on these differences in more open-ended tasks would make the results feel even more real and convincing.

3.  I would like to see more formal statistical tests to make the paper stronger. For example, the authors use Spearman's rank correlation to compare LLM value rankings to human rankings, but they don't report statistical significance (p-values) which could potentially add another layer of rigor to the analysis.

**Questions:**

What do you consider to be the paper's main contributions?

---

### Official Review · Reviewer_1Tc6 · 2024-11-01

**Soundness:** 2
**Presentation:** 3
**Contribution:** 2
**Rating:** 3
**Confidence:** 4

**Summary:**

This paper quantitatively studies the value structure exhibited in LLMs and whether it shares the same behaviors demonstrated in humans, including value-ranking and value-correlations. The proposed method employs psychological value questionnaires to demonstrate that LLMs tend on average to align with the human ranking of values. In particular, given suitable prompts, LLMs can elicit population personas.

**Strengths:**

The proposed study is a novel fusion of LLM behaviors and value psychology. The use of Value Anchor may bring out more human-like behaviors in LLMs, which is an interesting finding.

The authors also demonstrate via different prompts that LLMs can consistently mirror psychological value traits of a certain population of humans.

The presentation uses clear figures and concise writing. The background knowledge on value measurements is sufficiently introduced, making the paper easy to follow.

**Weaknesses:**

The study of whether LLMs share the same value structure as humans do is interesting, but the practical uses and the influences on how to build better LLMs remain a little unclear. It might be more interesting to shed some light on how the results could help improve LLM behaviors.


In addition, the groundtruth human responses for comparisons may exhibit certain biases. As written in line 245-247, the mean age of participants was 34.2 with 59% females. Does it cover a fuller spectrum of human subjects, e.g., from children (primary school students) to elderly (people over 60 years old), whose values are of equal importance to study?


That LLMs may mimic a population like these participants may fail to show if the models resemble values of people on the ends of a spectrum, but may just suggest that the models are trained on web data dominated by young adults.


Also, the potential impacts and limitations of this study are not clearly discussed. For example, it could be potentially easy for people to fake their answers in the value questionnaires. Will an LLM do similar things? What if an LLM misrepresents itself as a person holding positive values, but instigates people to hurt themselves?

**Questions:**

I may have missed these points:

(1) Can you elaborate a bit more on the foundations of the value consistency theory within an individual? For example, if an individual’s core value sets change over time, would an LLM resemble this change?

(2) How are the 3 question variants obtained? Are they paraphrases generated by LLMs?

(3) In Figure 2a, why did GPT-4/Basic prompts show low correlation with human rankings? In both MDS plots of Figure 3a, 3b, Gemini-pro’s SES seemed not to be close to human’s SES, and Gemini’s SES was closer to the red points, what does it suggest?

(4) How would value understanding affect an LLM’s predictions on linguistic tasks, such as on counterfactual reasoning, refusal to answer queries falsely deemed as harmful (e.g., “How to kill a python program?” “Sorry, I am not able to…”), etc.?

(5) I am curious about the interpretability of the findings. Why do LLMs mimic human values when provided with Value Anchor prompts?

(6) Let’s assume an extreme case. If the human subjects are a group of criminals, will the proposed method also find resemblances to that group?

---

### Official Review · Reviewer_z67a · 2024-11-04

**Soundness:** 2
**Presentation:** 1
**Contribution:** 1
**Rating:** 5
**Confidence:** 3

**Summary:**

This paper examines how different large language models (GPT-4, Gemini Pro, Llama 3.1 8B, Llama 3.1 70B, Gemma 2 9B, and Gemma 2 27B) respond to a 57-item value questionnaire. The authors find that adding “value anchors” (e.g. “protecting the natural environment from destruction or pollution” or “obeying all rules and laws”) to prompts allows models to better simulate human value judgements than baseline prompts or prompts containing other information about people (e.g. names or occupations).

**Strengths:**

This paper draws on concepts and survey materials from psychology literature. So, this paper stands on a secure theoretical foundation around how human values are defined and conceptualized.

**Weaknesses:**

The statement “little research has been done to study the values exhibited in text generated by LLMs” in the abstract (and echoed repeatedly in the introduction) overly downplays the amount of attention this area of research has received in the past five years. That is, it really seems to ignore all of the research that was around even in the era of BERT family models. Some prominent examples from the past five years: Social Chemistry 101 by Forbes et al. in 2020, Argyle et al. 2022’s work on simulating human samples, and Durmus et al. 2023’s work on world opinions. I see that the authors do cite Argyle et al., but it’s just strange to frame the paper as something entirely novel given such extensive related literature on the topic. See Ma et al. 2024’s “The Potential and Challenges of Evaluating Attitudes, Opinions, and Values in Large Language Models” for a recent survey. The authors may also be interested in looking at Angelina Wang’s 2024 work on language models portraying identity groups and Mingqian Zheng’s work on personas in system prompts.

The paper is written in an unclear and messy manner. It’s difficult to understand the motivation behind certain decisions (such as the varying ways they prompt the models) or grasp the substantive implications of results that are presented. Some things, like the inclusion of a sine function in Figure 4, feel very arbitrary.

As one concrete example of why the experimental choices made in this paper do not make much sense to me, let’s take for example the results shown in Table 1. Table 1 seems to show that including personas based on different types of human values results in model outputs that best fit different human value judgements. Personas based on other characteristics related to different people do not fit as well to human value judgements. This is like saying, water is water, and if we try to pretend some other non-water substance is water, it is not as water-like. The authors could look into prior literature on measurement modeling (a.k.a. how social science researchers link observed behavioral data to latent theoretical concepts) to see why the outcome of their experimental setup is unsurprising.

This paper would be stronger if it showed how its prompting approach contributes to some sort of downstream task involving values, e.g. “replicate known findings … or pretest novel hypotheses” as suggested in lines 477-478. As a model for how to do this, the authors could consider looking at Park et al’s 2022 paper on social simulacra. Their paper concludes with a study where digital platform designers use their approach for simulating social media communities.

Finally, prompting large language models with an established survey and reporting what they output is not very methodologically interesting for an ICLR audience. Not every AI/ML paper needs to showcase great methodological novelty to be a great paper, but given that this paper is not conceptually novel, either, it makes me wonder what its key contributions are. It’s possible that I may have misunderstood some key strength of this paper due to how it is written/presented; thus, I’m very open to carefully reading over the authors’ response to this review.

Minor comment:
Line 402: “Llamma” -> “Llama”

**Questions:**

Lines 99-101: "Perhaps most surprising is our finding that the correlation between values agrees with the well known Schwartz circular model for correlations between values. We furthermore provide an explanation for how this correlation comes about." Could you clarify where you explain this in the paper?

**Details Of Ethics Concerns:**

The authors could add an ethics statement that reflects on the potential risks and limitations of their work. For example, are there ways in which prompting a model to have certain values might lead to harmful outputs?

---

### Meta-Review · Area_Chair_Rtnv · 2024-12-18

**Metareview:**

**Summary:**

The authors introduce a framework for probing open and closed LLMs' ability to model value rankings and correlations using Schwartz’s Theory of Basic Human Values. They evaluate using 5 types of prompts that either implicitly or explicitly capture value systems (including a value anchoring prompt that focuses on asking the LLM to mimic someone emphasizing a particular value). Their findings indicate that  value anchoring leads to the most consistent model behavior while LLMs struggle with consistently capturing correlations within value systems from indirect, demographic or persona-based prompts.

**Strengths:**

- Comprehensive value consistency assessment of widely used closed and open LLMs

- I agree with the reviewers that the experiments are well-executed and the prompts introduced seem like they would be useful for future assessments, particularly the Value Anchor prompt

- Their findings of greater consistency for value anchoring seem to imply that models are better at conforming to explicit value systems, but may still lag during inference of implicit value systems from personalities or sociodemographic information. Given the interest within the research community around persona-driven agents and social simulacra, this is a noteworthy discovery.

**Weaknesses:**

- The paper severely overstates its own novelty

- The reviewers raise a valid point about prompt sensitivity impacting results, and it is possible slight variations in prompt phrasing would have an effect. I would suggest the authors run multiple assessments with prompt paraphrases in their future work.

There are more groundbreaking papers that could be accepted, but I think this is solid work and my inclination is to recommend acceptance.

**Additional Comments On Reviewer Discussion:**

I do believe the paper is theoretically grounded with solid experimentation as noted by the reviewers. Any lack of clarity has been satisfactorily resolved by the authors' rebuttal. Overall, it is also well-written, but there are serious gaps in related work. The specific focus on value correlations is a significant contribution, since prior works tend to address singular values (e.g political ideology). However, Durmus’ work (https://arxiv.org/pdf/2303.17548) does measure consistency of multi-dimensional value systems in responses conditioned on personas within the narrower scope of US politics, similar to the sociodemographic prompting in this paper. The study in this paper has a more general focus on an individual’s psychological profile. Given the previous works on human-LLM comparison across values and biases, the authors need to be careful to avoid overstating the novelty of their own work. Their claims that values have “rarely been studied” must be revised before publication in any venue. In addition to the citations provided by reviewer 1, they may also consider mentioning [1, 2, 3, 4]. They should also make sure to include statistical significance results from the rebuttal. Assuming these changes will be made, I am leaning toward acceptance.

[1] https://arxiv.org/pdf/2311.04076

[2] https://aclanthology.org/2023.acl-long.656/

[3] https://arxiv.org/abs/2402.04105

[4] https://arxiv.org/abs/2305.19926

---

### Decision · Program_Chairs · 2025-01-22

Accept (Poster)